# Critical Thermal Limits Do Not Vary between Wild-caught and Captive-bred Tadpoles of *Agalychnis spurrelli* (Anura: Hylidae)

**Pol Pintanel** [1,2], **Miguel Tejedo** [1], **Freddy Almeida-Reinoso** [2], **Andrés Merino-Viteri** [2,*] and **Luis Miguel Gutiérrez-Pesquera** [1,*]

1   Department of Evolutionary Ecology, Estación Biológica de Doñana, CSIC, Av. Américo Vespucio s/n, 41092 Sevilla, Spain; polpintanel@gmail.com (P.P.); tejedo@ebd.csic.es (M.T.)
2   Laboratorio de Ecofisiología and Museo de Zoología (QCAZ), Escuela de Ciencias Biológicas, Pontificia Universidad Católica del Ecuador, Av. 12 de Octubre y Roca, Aptdo., Quito 17-01-2184, Ecuador; fmalmeidaec@yahoo.com
*   Correspondence: armerino@puce.edu.ec (A.M.-V.); lmpesque@gmail.com (L.M.G.-P.)

**Abstract:** Captive-bred organisms are widely used in ecology, evolution and conservation research, especially in scenarios where natural populations are scarce or at risk of extinction. Yet, it is still unclear whether captivity may alter thermal tolerances, crucial traits to predict species resilience to global warming. Here, we study whether captive-bred tadpoles of the gliding treefrog (*Agalychnis spurrelli*) show different thermal tolerances than wild-caught individuals. Our results show that there are no differences between critical thermal limits ($CT_{max}$ and $CT_{min}$) of captive-bred and wild-caught tadpoles exposed to three-day acclimatization at 20 °C. Therefore, we suggest that the use of captive-bred amphibians is valid and may be appropriate in experimental comparisons to thermal physiological studies of wild populations.

**Keywords:** amphibians; $CT_{max}$; $CT_{min}$; Chocó; Ecuador; laboratory; microclimate; thermal tolerance

## 1. Introduction

Understanding how critical thermal limits in ectotherms vary through space and time has become of central importance in studies of ecology and evolution [1,2]. With the increasing threat of climate change, it has become essential to predict species persistence to increasing mean temperatures and episodic heat waves (e.g., [3–6]). However, most macro-ecological studies assume invariability in thermal limits within species and do not consider, for example, changes arising during ontogeny, space and time (e.g., [7–11]).

The methodology employed to estimate critical thermal limits may also lead to different estimates. For example, variation in the rate of heating/cooling [9,12], acclimation period [13], temperature during acclimation period [14], the photoperiod [15], the chosen end-point [16,17] or even whether experimental organisms were obtained from field or laboratory [18,19]. Sources of intraspecific variation arising from the captive environment are an important consideration in experimental biology, since many ecological, evolutionary and conservation studies on thermal tolerances have employed captive-bred species (e.g., [20–22]).

A major concern in ecological and evolutionary studies using laboratory-reared organisms is whether captive conditions may have altered ecological and physiological estimates [19] even when populations have been in laboratory conditions for only a few generations [23]. Captivity may affect organisms in different ways. For instance, populations kept in laboratory conditions for many

generations may have modified thermal tolerances through founder effects, genetic drift, inbreeding and evolutionary processes [18,19,24,25]. However, in populations maintained in captivity for only a few generations, other non-adaptive processes such as environmental conditions experienced in previous stages can influence thermal tolerance estimates on further developmental stages within a generation [26–29] or even across the next generation (e.g., via maternal effects; [30]).

Here, we tested whether the larvae of a population of the gliding treefrog (*Agalychnis spurrelli* Boulenger, 1931) maintained in laboratory conditions for one generation showed similar estimates in heat and cold tolerance (critical thermal limits; $CT_{max}$ and $CT_{min}$) to wild-caught larvae originating from the same field site. We hypothesize that if captive-bred and wild-caught individuals are not fundamentally different, the use of captive-bred populations of some species in physiological and ecological studies might be representative of wild populations.

## 2. Materials and Methods

### 2.1. Study Populations and Thermal Variability

*Agalychnis spurrelli* is a tropical arboreal frog, presenting an aquatic larval stage, found below 850 metres of elevation in Central America and the north of South America [31]. We collected *A. spurrelli* tadpoles (*n* = 40; Gosner stage 26–37; Gosner 1967) from a population in Durango, in the northern coast of Ecuador in the Chocoan lowland rainforest region (1°02′ N, 78°37′ W; 227 m a.s.l.) on June 2014. We also obtained first generation (F1) *A. spurrelli* tadpoles (*n* = 32; Gosner stage 26-27; Gosner 1967) from frogs originating from the same field locality, and kept in the "Balsa de los Sapos" Amphibian Conservation Initiative facilities in the Pontificia Universidad Católica del Ecuador (PUCE). Captive-bred tadpoles came from two clutches from two different pairs of parents. Individuals from the two clutches were then mixed at random. We placed each tadpoles group, captive-bred and field-caught, in a plastic container filled with 4 L of dechlorinated water and maintained at a constant temperature of 20 °C with a photoperiod of 12L:12D. Tadpoles were fed rabbit chow *ad libitum* three times a week post full water change.

In order to characterize the temperatures to which individuals are exposed, we deployed HOBO Pendant 64K temperature dataloggers to obtain continuous recordings of both air and water temperatures from the field (every 15 min). Dataloggers were attached to a branch (approx. 1 meter above the ground) and to the bottom of the permanent pond where tadpoles were captured to record air and water temperatures. Finally, we extracted the maximum and the minimum temperatures of dataloggers for each day.

### 2.2. Estimates of Critical Thermal Limits

Before conducting thermal tolerance assays, animals were kept at laboratory conditions for at least three days. This period was chosen to stabilize $CT_{max}$ and $CT_{min}$ after a large change in acclimation temperature to field or laboratory conditions [13,32]. To estimate critical thermal limits ($CT_{max}$ and $CT_{min}$), we followed the dynamic method of Lutterschmidt and Hutchison [33]. We randomly assigned field and laboratory tadpoles to estimate $CT_{max}$ (24 field and 16 lab) and $CT_{min}$ (16 field and 16 lab). We placed each individual in a 100 mL plastic container at a starting temperature of 20 °C within a 15 L HUBER K15-cc-NR bath (Kältemaschinenbau AG, Germany) that was heated or cooled at a constant rate of 0.25 °C/min. The end-point was signalled for both thermal limits as the point at which the tadpole's mobility ceases completely and failed to respond to external stimuli. At that point, we measured water temperature with a Miller & Weber quick-recording thermometer (0.1 °C accuracy) placed beside the tadpole. We assumed that body temperature equalled water temperature because of tadpoles' small size [16,32]. Then we transferred the tadpoles to a plastic cup with water at the initial temperature, allowing for recovery within two hours. After the test, each tadpole was assigned a Gosner stage [34] and weighed to the nearest 0.001 g. We excluded tadpoles over Gosner stage

37 because metamorphic individuals tend to show reduced thermal tolerances [7,27]. Each individual was tested only once for either $CT_{max}$ or $CT_{min}$ (see Supplementary dataset 1 for raw data).

### 2.3. Statistical Analyses

We employed a multi-model inference approach using the "lm" function in R to examine the relationships between the critical thermal limits (dependent variable) and both size (i.e. body mass) and population (field or laboratory). We selected the best model using Akaike information criterion (AIC), Bayesian information criterion (BIC) and Log-likelihood ratio tests. All analyses were conducted in R v.3.6.1 [35].

## 3. Results

We found no difference in critical thermal limits between wild-caught tadpoles and captive-bred tadpoles of *A. spurrelli*. Specifically, wild-caught and captive-bred tadpoles exhibited similar $CT_{max}$ whilst wild-caught and captive-bred tadpoles exhibited similar $CT_{min}$ (Figure 1; Table 1). Wild-caught tadpoles had bigger sizes than captive-bred individuals ($F_{1,70} = 379.9$, $p < 0.001$).

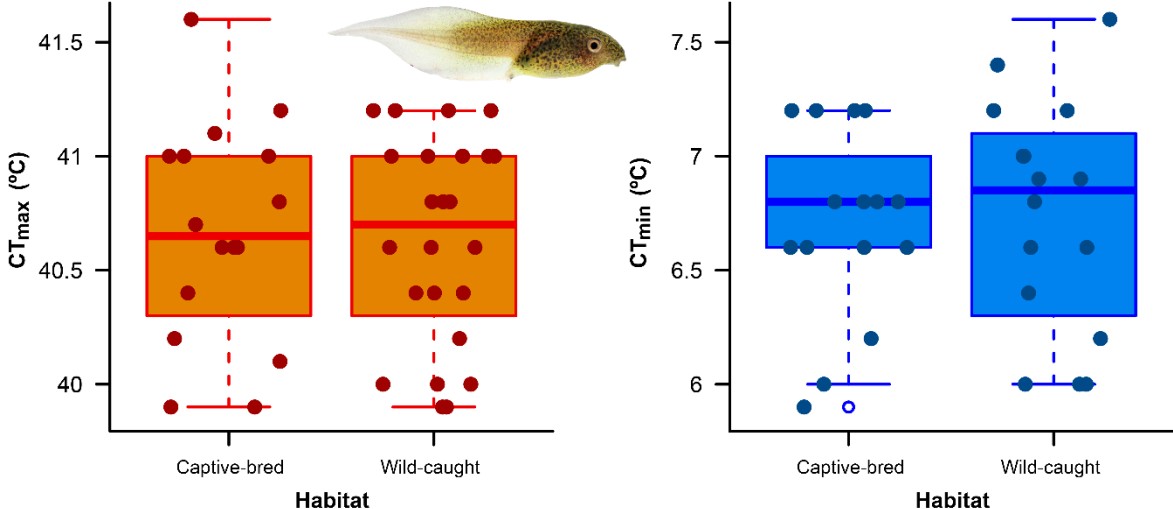

**Figure 1.** Variation in critical thermal maximum ($CT_{max}$) and critical thermal minimum ($CT_{min}$) between captive-bred and wild-caught tadpoles of *Agalychnis spurreli* from Durango, Ecuador. The boxplots show first and third quartiles (boxes), the median (line in box) and the minimum and maximum values (whiskers, excluding outliers).

**Table 1.** Models to test the effects of captivity (population) and weight in the critical thermal limits, $CT_{max}$ and $CT_{min}$, of *A. spurrelli* according to Akaike information criterion (AIC), Bayesian information criterion (BIC) and Log-likelihood (LogLik) tests. Log-likelihood ratio tests are done always against the best model in order to test the significance of each term. Best models are marked with an asterisk.

| Critical Thermal Limit | Model | df | AIC | BIC | LogLik | *p*-Value |
|---|---|---|---|---|---|---|
| $CT_{max}$ | * $CT_{max}$ ~ 1 | 2 | 54.026 | 57.404 | −25.013 | |
| | $CT_{max}$ ~ mass | 3 | 55.991 | 61.058 | −24.996 | 0.855 |
| | $CT_{max}$ ~ population | 3 | 55.967 | 61.034 | −24.984 | 0.813 |
| | $CT_{max}$ ~ mass + population | 4 | 57.953 | 64.708 | −24.976 | 0.967 |
| | $CT_{max}$ ~ mass * population | 5 | 59.581 | 68.026 | −24.791 | 0.940 |
| $CT_{min}$ | * $CT_{min}$ ~ 1 | 2 | 43.083 | 46.014 | −19.541 | |
| | $CT_{min}$ ~ mass | 3 | 44.747 | 49.144 | −19.373 | 0.574 |
| | $CT_{min}$ ~ population | 3 | 45.043 | 49.441 | −19.522 | 0.848 |
| | $CT_{min}$ ~ mass + population | 4 | 46.203 | 52.066 | −19.102 | 0.668 |
| | $CT_{min}$ ~ mass * population | 5 | 44.862 | 52.191 | −17.431 | 0.267 |

Temperatures from Durango did not exhibit seasonal patterns; total thermal variation is largely driven by daily temperature variation. Field air temperatures are more extreme than water temperatures. Maximum air temperatures were higher than water temperatures. Minimum air temperatures were also lower than water temperatures (Figure 2).

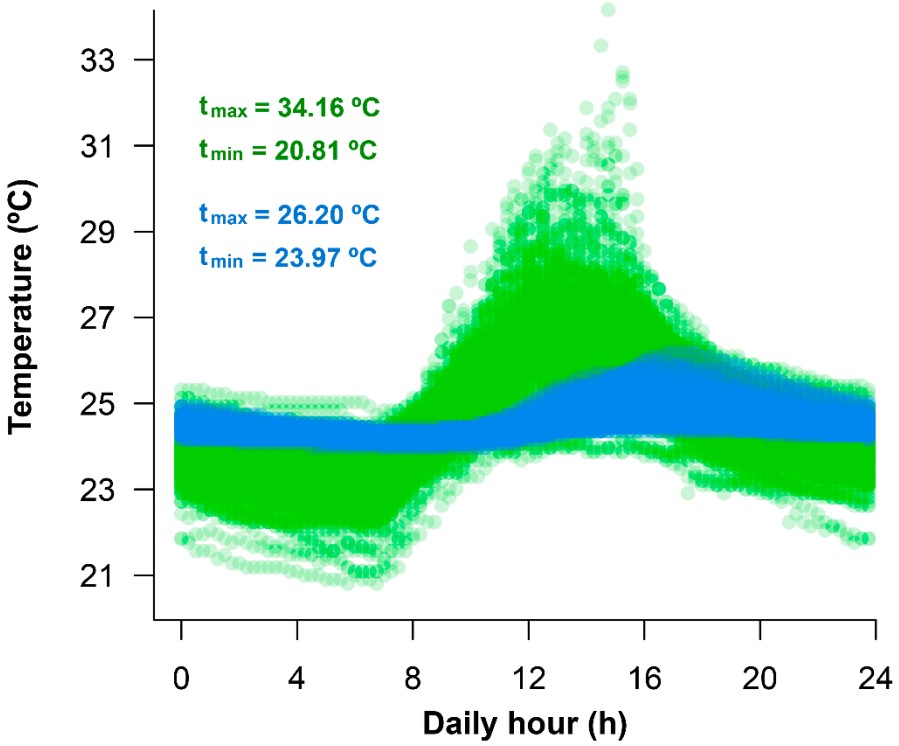

**Figure 2.** Profiles of daily variation of air (green; *n* = 241 days) and pond water (blue; *n* = 115 days) temperature in a natural habitat of *A. spurrelli* in Durango (Ecuador).

## 4. Discussion

Understanding how thermal tolerance varies in response to captivity is needed to better assess the vulnerability of species to climate change. Our results demonstrate that thermal tolerances of *A. spurrelli* tadpoles do not vary between wild-caught and captive-bred animals. Previous studies comparing wild and laboratory lines of ectotherms revealed contrasting patterns in thermal tolerances. Fishes showed either higher [36,37] or lower [18] thermal tolerance in domesticated in contrast to wild animals. However, those studies did not provide an acclimatization period before the experiments were carried on. Therefore, these differences may be biased to an acclimation to previously exposed temperatures [13] rather than differences between wild/captive populations. For instance, Morgan et al. [18] found no differences between wild-caught and captive-bred zebrafish acclimatized under laboratory conditions. Another possible example is found in Krebs et al. [38] where they found an increase in thermal tolerance just after transferring wild fruit flies to a laboratory medium, but then thermal tolerance remained constant even after more than 50 generations. Therefore, critical thermal limits may be less labile to laboratory conditions than previously thought (also see [19]). For instance, Enriquez-Urzelai et al. [27] found that thermal tolerances in juvenile frogs of *Rana temporaria* were not modified by previous larval developmental acclimation temperatures. However, the proposed invariability in thermal tolerance estimates in amphibians exposed to laboratory conditions requires further testing including species from different elevations and with broader differences between laboratory and field thermal conditions. For instance, Jensen et al., [23] found strong differences in $CT_{max}$ and $CT_{min}$ in laboratory-raised first generation individuals when compared with field collected individuals in a collembolan. This study is more comparable to our results since they standardized thermal conditions prior to tolerance assays. Hence, we cannot disregard that captive frogs would exhibit similar thermal tolerance changes in

later generations in captivity. Some experiments demonstrate that thermal physiology can change in response to selection pressure applied under laboratory conditions [24,39–41].

Captive-bred populations may offer a variety of opportunities to conduct research that improves our knowledge on species ecology and evolution [42]. In some cases, the use of captive-bred populations is more practical and responsible when it is necessary to limit the impact upon wild populations. For instance, the "Balsa de los Sapos" Amphibian Conservation Initiative holds 17 species that are under a International Union for Conservation of Nature (IUCN) extinction risk category (vulnerable, endangered and critically endangered); therefore, in the rare event of finding wild individuals of these endangered species, their use for research may not be ethically sound. Conservation initiatives may benefit from integrating efforts along with research programs to develop applied solutions to facilitate the restoration of wild populations. For example, physiological data obtained from captive-bred populations may give a better assessment of populations' vulnerability to climate change that could ultimately lead to identifying locations where species may still persist for later reestablishment.

In *A. spurrelli*, given that pond temperature is significantly lower than tadpoles' heat tolerance, the thermal safety margin to acute stress (difference between heat tolerance and maximum environmental temperature) was very high, ranging from 13.7 °C to 15.4 °C. Thus, vulnerability risk to increasing temperatures of this lowland population is pretty low. On the other hand, this result implies another question: why is heat tolerance so high compared to environmental temperatures? First, water temperature was taken at the bottom of the pond at a depth of approximately half a meter, while the tadpoles were generally on the top of the water layer where temperatures are usually higher. Furthermore, physiological traits are subject to some intrinsic constraints [43]. For example, the "hotter is better" hypothesis predicts that species adapted to high temperatures will have higher performances than those adapted to colder temperatures [43,44], since adaptation is unable to overcome the depressant effects of low temperatures [45]. Thus, species may benefit from high heat tolerances even in cold environments.

We acknowledge that the use of only one population of a single species limits our ability to predict whether laboratory conditions would or would not show plastic or evolutionary responses to thermal physiological traits. Even though we are aware of the problems associated to single-species estimations in thermal tolerances (e.g., limitation in the generalization of effects of laboratory conditions, unknown interspecific variability), studies available on this topic are still scarce for vertebrates. While using more species/populations is ideal, a scarcity of captive-bred neotropical vertebrate populations as well as funding and time [42] make it a complex task. Thus, given the importance of improving the management of threatened species, we consider that the study is justified. Significant body of knowledge on present amphibian conservation strategies are based on captive single-species studies (e.g., [46–48]). Further studies should focus on expanding the available information to disentangle the effects of captivity in physiological traits estimates by, for instance, increasing the number of populations, species and generations spent in captivity.

Our results contribute to the understanding of the effects of laboratory conditions on the estimates of critical thermal limits. Since critical thermal limits obtained from captive-bred tadpoles of the gliding treefrog (*A. spurrelli*) do not differ from those obtained from field-captured individuals, the use of captive-bred species may be appropriate in some species to address questions about thermal physiological adaptation and assess the impacts of anthropogenic global warming on organisms. Integration of research goals and captive breeding programs has already improved collection management and sustainability [42]. This kind of basic research on vertebrate thermal physiology is crucial before we can jump to applied solutions. In the future, physiological data of captive-bred species could be used to create habitat suitability models that could ultimately lead to the reestablishment of populations in the wild.

**Supplementary Materials:** The following is available online at http://www.mdpi.com/1424-2818/12/2/43/s1, Supplementary dataset 1: Dataset with information about critical thermal limits of captive and field populations of *Agalychnis spurrelli*.

**Author Contributions:** Conceptualization, P.P., M.T. and L.M.G.-P.; methodology, P.P., M.T. and L.M.G.-P.; formal analysis, P.P.; field work, P.P., F.A.-R. and L.M.G.-P.; experiments, L.M.G.-P.; writing—review and editing, P.P., M.T., F.A.-R., A.M.-V. and L.M.G.-P.; funding acquisition, M.T. and A.M.-V. All authors have read and agreed to the published version of the manuscript.

**Funding:** This study was supported by AECID (AP/038788/11) and MINECO (CGL2012-40246-C02-01 and CGL2017-86924-P) grants to M.T. and A.M. and Severo Ochoa (SEV-69) funds to M.T. Frogs *ex* situ management was funded by Dirección General Académica of PUCE through research grant L13227 to A.M.

**Acknowledgments:** We thank the staff from the "Balsa de los Sapos" Amphibian Conservation Initiative in PUCE for helping in fieldwork and laboratory and providing the captive-bred individuals. Mayra Castro, Andrea López Rosero, Mari Piñero and David Velalcázar helped in fieldwork. Sofia Salinas-Ivanenko and Phil Jervis provided helpful comments. Ministerio del Ambiente of Ecuador provided the permits to conduct this investigation (005-14/003-15/012-015 IC-FAU-DNB/MA; 12-2015-FAU-DPAP-MA).

**Conflicts of Interest:** The authors declare no conflict of interest.

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
