# Peer review of "Critical Thermal Limits Do Not Vary between Wild-caught and Captive-bred Tadpoles of Agalychnis spurrelli (Anura: Hylidae)"

_diversity, doi:10.3390/d12020043_

Round 1

Reviewer 1 Report

This study provides empirical data comparing critical thermal traits between wild-caught and captive-bred tadpole of a Neotropical tree frog species, and develops our understanding of thermal ecology of ectotherms. The authors didn’t find a significant difference in tolerance to heat and cold between wild-caught vs. and captive-bred tadpoles, and suggested that the use of captive-bred individuals is acceptable for research focusing on thermal physiology of amphibians. The introduction, methods, and statistical analyses are appropriate to the scientific questions being addressed. However, I have two main concerns: the first one pertains to the way the hypotheses are presented and the second pertains to the conclusions made. I provide further comments on this below, but before I would like to comment on the merits of the Introduction.

I agree with the observation that "most macro-ecological studies assume invariability in thermal limits within species and do not consider, for example, changes arising during ontogeny, space and time. [e.g. 7,8-11]." (lines 28-30). This is important because it’s a common assumption that continues to be made in studies focusing on macro-ecology and species distribution modeling.

In addition to the studies cited by the authors (which focus on ontogeny), there are several recent studies that focus on natural populations of amphibians and other ecthotherms and that also provided strong evidence that within-population (& intra-specific) variability is an important attribute of species’ thermal physiology.

In the Introduction, the authors stated that they tested wether wild-caught tadpoles and captive-bred tadpoles of Agalychnis spurreli have similar heat and cold tolerance. From this, one can infer the following Null vs. Alternative Hypotheses:

Tolerance to heat (CTmax):

Null Hypothesis: wild-caught tadpoles and captive-bred tadpoles do not differ in their tolerance to heat

Alternative Hypothesis: wild-caught tadpoles and captive-bred tadpoles differ in their tolerance to heat

Tolerance to cold (CTmin):

Null Hypothesis: wild-caught tadpoles and captive-bred tadpoles do not differ in their tolerance to heat

Alternative Hypothesis: wild-caught tadpoles and captive-bred tadpoles differ in their tolerance to heat

In the Results, the authors report that they "found strong statistical support for the hypothesis that wild-caught tadpoles of A. spurreli did not differ significantly from captive-bred tadpoles in CTmax or CTmin..." (lines 92-93).

I suggest revising and rewriting this statement. Strictly speaking, a statistical test should focus on the null hypothesis. We use hypothesis testing to decide if the data are compatible with the null hypothesis. To do this, we calculate the probability of a mismatch that is as extreme or more extreme than that observed (data). To calculate this probability, we determine the sampling distribution of the test statistic assuming that the null hypothesis is true. (The sampling distribution is automatically computed in most statistical packages.) Following the test, you either reject or fail to reject the null hypothesis.

Therefore, a suggested updated text could be:

"We failed to reject the null hypothesis that wild-caught tadpoles of A. spurreli did not differ significantly from captive-bred tadpoles in CTmax or CTmin..."

Or, put in more accessible terms: "We found no difference in critical thermal traits between wild-caught tadpoles and captive-bred tadpoles of A. spurreli. Specifically, wild-caught and captive-bred tadpoles exhibited similar CTmax whilst wild-caught and captive-bred tadpoles exhibited similar CTmin."

Additionally, the authors need to clarify what they mean by "alternative hypotheses" when reporting the results (lines 93-94):

Following the opening statement in the Results (lines 92-93), it is clear to the reader that they are referring to the null hypothesis as formulated above (see Null vs. Alternative Hypotheses). From this, one would understand that the alternative hypotheses is that "wild-caught tadpoles and captive-bred tadpoles *differ* in their tolerance to heat (or cold)"

However, the authors make a comparison with "against the alternative hypotheses that critical thermal limits differed among populations and/or body size (Table 1)," which is different than the alternative hypotheses that one would infer based on the null hypothesis of no difference of tolerance to heat and cold. Please clarify.

My second concern pertains to the conclusions being made. While the results clearly indicate that there is no significant difference in tolerance to heat and cold between wild-caught vs. and captive-bred tadpoles, the authors should acknowledge that the two groups of tadpoles experienced relatively similar thermal conditions prior to the experiment.

Captive captive-bred tadpoles were kept at laboratory conditions at a constant temperature of 20°C with a balanced photoperiod (12L:12D), whereas wild-caught tadpoles experienced conditions at relative constant temperatures between 24 and 26°C. Given that the difference in thermal regimes between the two groups is relatively small (4-6°C) and, importantly, both are relatively “mesic” temperatures, the conclusion that using captive-bred individuals in thermal physiology studies is acceptable yet limited to a narrow thermal range of 20 and 26°C. Nevertheless, the findings are contrasted with finding from previous studies in different groups organisms and the authors raise an important possibility: that “critical thermal limits may be less labile to laboratory conditions than previously expected.” This obviously requires further testing, especially in species that experience more extreme thermal regimes (e.g., high-elevation taxa that experience lower temperatures, e.g. 5, 10, 15°C).

It’s good to see that the authors acknowledged the limitations of the study, namely the use of only one population of a single species limits our ability to make broader predictions. Nevertheless, the findings of this study will motivate further work on this topic. I look forward to seeing the supplementary materials. Additionally, it would be great if the authors share the raw data so the analyses can be replicated and compared with similar data on thermal physiology of amphibian larvae.

Additional corrections:

In Table 1, replace "weight" by "mass"

Author Response

Response to Reviewer 1 Comments

Authors (Au.) -> In the revised version we have included the comments and suggestions proposed. Thank you for the effort put in the manuscript.

Comments and Suggestions for Authors

This study provides empirical data comparing critical thermal traits between wild-caught and captive-bred tadpole of a Neotropical tree frog species, and develops our understanding of thermal ecology of ectotherms. The authors didn’t find a significant difference in tolerance to heat and cold between wild-caught vs. and captive-bred tadpoles, and suggested that the use of captive-bred individuals is acceptable for research focusing on thermal physiology of amphibians. The introduction, methods, and statistical analyses are appropriate to the scientific questions being addressed. However, I have two main concerns: the first one pertains to the way the hypotheses are presented and the second pertains to the conclusions made. I provide further comments on this below, but before I would like to comment on the merits of the Introduction.

 I agree with the observation that "most macro-ecological studies assume invariability in thermal limits within species and do not consider, for example, changes arising during ontogeny, space and time. [e.g. 7,8-11]." (lines 28-30). This is important because it’s a common assumption that continues to be made in studies focusing on macro-ecology and species distribution modeling.

 In addition to the studies cited by the authors (which focus on ontogeny), there are several recent studies that focus on natural populations of amphibians and other ecthotherms and that also provided strong evidence that within-population (& intra-specific) variability is an important attribute of species’ thermal physiology.

 In the Introduction, the authors stated that they tested wether wild-caught tadpoles and captive-bred tadpoles of Agalychnis spurreli have similar heat and cold tolerance. From this, one can infer the following Null vs. Alternative Hypotheses:

 Tolerance to heat (CTmax):

 Null Hypothesis: wild-caught tadpoles and captive-bred tadpoles do not differ in their tolerance to heat

 Alternative Hypothesis: wild-caught tadpoles and captive-bred tadpoles differ in their tolerance to heat

 Tolerance to cold (CTmin):

 Null Hypothesis: wild-caught tadpoles and captive-bred tadpoles do not differ in their tolerance to heat

 Alternative Hypothesis: wild-caught tadpoles and captive-bred tadpoles differ in their tolerance to heat

 In the Results, the authors report that they "found strong statistical support for the hypothesis that wild-caught tadpoles of A. spurreli did not differ significantly from captive-bred tadpoles in CTmax or CTmin..." (lines 92-93).

 I suggest revising and rewriting this statement. Strictly speaking, a statistical test should focus on the null hypothesis. We use hypothesis testing to decide if the data are compatible with the null hypothesis. To do this, we calculate the probability of a mismatch that is as extreme or more extreme than that observed (data). To calculate this probability, we determine the sampling distribution of the test statistic assuming that the null hypothesis is true. (The sampling distribution is automatically computed in most statistical packages.) Following the test, you either reject or fail to reject the null hypothesis.

 Therefore, a suggested updated text could be:

 "We failed to reject the null hypothesis that wild-caught tadpoles of A. spurreli did not differ significantly from captive-bred tadpoles in CTmax or CTmin..."

 Or, put in more accessible terms: "We found no difference in critical thermal traits between wild-caught tadpoles and captive-bred tadpoles of A. spurreli. Specifically, wild-caught and captive-bred tadpoles exhibited similar CTmax whilst wild-caught and captive-bred tadpoles exhibited similar CTmin."

 Additionally, the authors need to clarify what they mean by "alternative hypotheses" when reporting the results (lines 93-94):

 Following the opening statement in the Results (lines 92-93), it is clear to the reader that they are referring to the null hypothesis as formulated above (see Null vs. Alternative Hypotheses). From this, one would understand that the alternative hypotheses is that "wild-caught tadpoles and captive-bred tadpoles *differ* in their tolerance to heat (or cold)"

 However, the authors make a comparison with "against the alternative hypotheses that critical thermal limits differed among populations and/or body size (Table 1)," which is different than the alternative hypotheses that one would infer based on the null hypothesis of no difference of tolerance to heat and cold. Please clarify.

Au. -> We agree. Our statement was incorrect. We made the changes proposed.

 My second concern pertains to the conclusions being made. While the results clearly indicate that there is no significant difference in tolerance to heat and cold between wild-caught vs. and captive-bred tadpoles, the authors should acknowledge that the two groups of tadpoles experienced relatively similar thermal conditions prior to the experiment.

 Captive captive-bred tadpoles were kept at laboratory conditions at a constant temperature of 20°C with a balanced photoperiod (12L:12D), whereas wild-caught tadpoles experienced conditions at relative constant temperatures between 24 and 26°C. Given that the difference in thermal regimes between the two groups is relatively small (4-6°C) and, importantly, both are relatively “mesic” temperatures, the conclusion that using captive-bred individuals in thermal physiology studies is acceptable yet limited to a narrow thermal range of 20 and 26°C. Nevertheless, the findings are contrasted with finding from previous studies in different groups organisms and the authors raise an important possibility: that “critical thermal limits may be less labile to laboratory conditions than previously expected.” This obviously requires further testing, especially in species that experience more extreme thermal regimes (e.g., high-elevation taxa that experience lower temperatures, e.g. 5, 10, 15°C).

Au. -> We agree that both lab and field thermal environments are relatively similar thus we could expect that thermal lab selection, if it were to happen, could be of a minor magnitude. But a fast response to selection (R) depends basically not only on selection differential, that will presumably be higher at field environments with extreme values than that expected in milder lab conditions, but also R depends on the amount of additive variance in CTmax and CTmin that is not known in our species but is generally considered of small magnitude in other ectotherms: Drosophila sp. (Diamond 2016) and fishes (Baer and Travis, 2000; Doyle et al., 2011; Meffe et al., 2011). So, any conclusion on this subject is very speculative and would require hard work. We believe that this is not the focus of the article but have added the following sentence addressing this aspect: “However, the proposed invariability in thermal tolerance estimates in amphibians exposed to laboratory conditions requires further testing including species from different elevations and with broader differences between laboratory and field thermal conditions. For instance, Jensen et al., [23] found strong differences in CTmax and CTmin in laboratory raised first generation individuals when compared with field collected individuals in a collembolan. This study is more comparable to our results since they standardized thermal conditions prior to tolerance assays. Hence, we cannot disregard that captive frogs would exhibit similar thermal tolerance changes in later generations in captivity”.

 It’s good to see that the authors acknowledged the limitations of the study, namely the use of only one population of a single species limits our ability to make broader predictions. Nevertheless, the findings of this study will motivate further work on this topic. I look forward to seeing the supplementary materials. Additionally, it would be great if the authors share the raw data so the analyses can be replicated and compared with similar data on thermal physiology of amphibian larvae.

Au. -> We provided the raw data in supplementary materials.

 Additional corrections:

 In Table 1, replace "weight" by "mass"

Au. -> Done

References

Baer, C. F.; Travis, J. Direct and correlated responses to artificial selection on acute thermal stress tolerance in a livebearing fish. Evolution 2000, 54, 238–244.

Doyle, C. M.; Leberg, P. L.; Klerks, P. L. Heritability of heat tolerance in a small livebearing fish, Heterandria formosa. Ecotoxicology 2011 20, 535–542.

Meffe, G. K.; Weeks, S. C.; Mulvey, M.; Kandl, K. L. Genetic differences in thermal tolerance of eastern mosqyitofish (Gambusia holbrooki; Poeciliidae) from ambient and thermal ponds. Can. J. Fish. Aquat. Sci. 2011, 52, 2704–2711.

Reviewer 2 Report

The present study compares the critical temperature limits tolerated by the tadpoles of a treefrog species, when taken directly from the field or the F1 of a captive-bred population. The goal of the study is to check if captive individuals offer the same results as those from field samples. I recognize the methodological value of this objective, but I have a number of important concerns regarding how the study has been carried out:

1.- No details are given on the conditions of captivity of treefrogs in the ““Balsa de los Sapos” Amphibian Conservation Initiative facilities. If the conditions are very similar to those found in the wild, it is normal for the results to be similar. The temperature of the wild zone is offered, but not of the zone of captivity. The objective of presenting only the temperature of the wild zone is not clear. Ideally, you must have the same data from both areas and compare them. Without an appropriate comparison between the captivity and the wild zone, the study is not publishable.

2.- A sample size of 72 tadpoles is used, but it is not reported whether they came from the same clutches or not. If 32-40 tadpoles are used per population, but all come from 2-3 clutches, that is, they are sibs, there is a severe pseudoreplication problem that would seriously affect the results. If they come from 10-15 clutches, there would be some sibs, but the problem could be solved using mixed models, with clutch as random factor. This point is important to know if the results are valid.

3.- F1 is compared with the wild population. Although this is valuable in itself, I have my doubts about whether the results would remain similar in case of comparing later generations. I also have my doubts about what would happen if the acclimatization step is ignored, which is perhaps the one that causes the homogenization of results. For the study to be sound, the authors should make more comparatives: what happens if acclimatization is not done? What happens if the number of generations in captivity increase? The study seems to be preliminary and more work is needed to be a solid study.

4.- Since there are no significant differences, it is important to know if these are due to a low effect size or a low sample size. The authors should calculate the statistical power and perform a finer analysis of their results to conclude that there are no statistical differences because there are no real differences.

5.- The Gosner stadium is measured, but then it is not included in the analyses. It should be introduced as a predictor in the models. In addition, it would be important to know that the tadpoles used in each treatment do not differ in size or in Gosner stage. These analyses are omitted in the typescript and have their importance, even if these factors are introduced into the models.

Minor comments:

L. 59: What do you mean with “the Chocó biogeographic hotspot”. At the least, a reference supporting the study area is a hotspot of biodiversity (if this is what you mean) is necessary.

L. 116: It is not the same domesticated lines that F1 of field captured individuals. Domesticated lines imply a number of generations under selective pressures given in captivity. This might do the difference.

L. 129: “that thermal physiology can increase”, this sentence has not sense. Thermal physiology may vary, but not increase or decrease. Thermal critical limits may increase or decrease. The sentence is vague and should be re-written.

L. 148: You say: “water temperature was taken at the bottom of the pond at a depth of approximately half a meter, while the tadpoles were generally on the top of the water layer where temperatures are usually higher.” This is awkwardness. If you know that tadpoles inhabit in the top of the water, why do you measure temperature at bottom?

Author Response

Response to Reviewer 1 Comments

Authors (Au.) -> In this revision we have responded to the comments provided. Thank you for the efforts put in the manuscript.

The present study compares the critical temperature limits tolerated by the tadpoles of a treefrog species, when taken directly from the field or the F1 of a captive-bred population. The goal of the study is to check if captive individuals offer the same results as those from field samples. I recognize the methodological value of this objective, but I have a number of important concerns regarding how the study has been carried out:

 1.- No details are given on the conditions of captivity of treefrogs in the “Balsa de los Sapos” Amphibian Conservation Initiative facilities. If the conditions are very similar to those found in the wild, it is normal for the results to be similar. The temperature of the wild zone is offered, but not of the zone of captivity. The objective of presenting only the temperature of the wild zone is not clear. Ideally, you must have the same data from both areas and compare them. Without an appropriate comparison between the captivity and the wild zone, the study is not publishable.

Authors (Au.) -> We initially included air temperature of the laboratory where frogs were maintained; however, a previous reviewer mentioned that this information was not relevant to the overall study given that the target life stage is fully aquatic. We decided to maintain field temperatures since they are more informative for the analysis of warming tolerance. Tadpoles from the field and “Balsa de los Sapos” were maintained at constant 20°C room temperature. We changed L65-67: “We placed each tadpoles group, captive-bred and field-caught, in a plastic container filled with 4 l of dechlorinated water and maintained at a constant temperature of 20 °C with a photoperiod of 12L:12D”. While we agree with the previous reviewer, we can include the data if necessary.

 2.- A sample size of 72 tadpoles is used, but it is not reported whether they came from the same clutches or not. If 32-40 tadpoles are used per population, but all come from 2-3 clutches, that is, they are sibs, there is a severe pseudoreplication problem that would seriously affect the results. If they come from 10-15 clutches, there would be some sibs, but the problem could be solved using mixed models, with clutch as random factor. This point is important to know if the results are valid.

Au. -> Since we captured field tadpoles rather than clutches we do not know whether they came from one or more clutches. Agalychnis spurrelli have explosive breeding events and thus, the field-captured tadpoles probably came from more than one clutch.

“Balsa de los Sapos” tadpoles were obtained from captive breeding of two different pairs of parents. We included this phrase L63:  “Captive-bred tadpoles came from two clutches from two different pairs of parents”. We agree that employing two clutches is a limitation in the study. However, the range of variation in thermal limits is similar to that found in field-collected tadpoles. There is extensive literature on the effects of captivity in Drosophila but only a few references in vertebrates, specifically fish. Obtaining the amount of clutches from different parents mentioned is almost impossible in some vertebrates. Amphibian conservation initiatives rarely host a captive population big enough to fulfil this requirement (see Lewis et al., 2019). We stressed this point in the second to last paragraph.

3.- F1 is compared with the wild population. Although this is valuable in itself, I have my doubts about whether the results would remain similar in case of comparing later generations. I also have my doubts about what would happen if the acclimatization step is ignored, which is perhaps the one that causes the homogenization of results. For the study to be sound, the authors should make more comparatives: what happens if acclimatization is not done? What happens if the number of generations in captivity increase? The study seems to be preliminary and more work is needed to be a solid study.

Au. -> All these suggestions are very interesting for future studies. We tried to acknowledge the limitations of the study in the second to last paragraph. We agree that a single generation in captivity may not be enough to drive a response to laboratory conditions. However, Jensen et al. (2019) reported differences in CTmax and CTmin in first-generation laboratory raised collembolan species (Orchesella cincta). We emphasized this in the discussion that our results are applicable only for the first generation L142-147: “Jensen et al., [23] found strong differences in CTmax and CTmin in laboratory raised first-generation individuals when compared with field-collected individuals in a collembolan. This study is more comparable to our results since they standardized thermal conditions prior to tolerance assays. Hence, we cannot disregard that captive frogs would exhibit similar thermal tolerance changes in later generations in captivity”.

On the other hand, we think that tadpoles coming from contrasting thermal environments (field and lab) express similar thermal tolerances in spite of (not because) us standardizing their previous thermal conditions before conducting the assays. The acclimatization step is necessary or else the results would correspond to a response to the previous thermal environment experienced by tadpoles (Brattstrom , 1968; Jensen et al., 2019).

 4.- Since there are no significant differences, it is important to know if these are due to a low effect size or a low sample size. The authors should calculate the statistical power and perform a finer analysis of their results to conclude that there are no statistical differences because there are no real differences.

Au. -> We agree. Our previous wording could lead to misunderstanding. We have changed the sentence to clarify that. L97-99: “We found no difference in critical thermal traits between wild-caught tadpoles and captive-bred tadpoles of A. spurrelli. Specifically, wild-caught and captive-bred tadpoles exhibited similar CTmax whilst wild-caught and captive-bred tadpoles exhibited similar CTmin

 5.- The Gosner stadium is measured, but then it is not included in the analyses. It should be introduced as a predictor in the models. In addition, it would be important to know that the tadpoles used in each treatment do not differ in size or in Gosner stage. These analyses are omitted in the typescript and have their importance, even if these factors are introduced into the models.

Au. -> We included the phrase: “Wild-caught tadpoles had bigger sizes than captive-bred individuals (F1,70 = 379.9, P < 0.001)”.

Gosner stage is correlated to size and is not a “normal” variable since some stages can last a few minutes while others may take days. For this reason, we think that size (i.e. mass) is a better variable for this analysis. Gosner stage was measured in order to exclude individuals over 38 Gosner stage (near the metamorphic climax), since tadpoles tend to have reduced thermal tolerances (Floyd, 1983 – Comp. Biochem. Physiol. A. 75: 267-271). We included this in L88-89: “We excluded tadpoles over Gosner stage 37 because metamorphic individuals tend to show reduced thermal tolerances [7,26]” to stress it.

 Minor comments:

L59: What do you mean with “the Chocó biogeographic hotspot”. At the least, a reference supporting the study area is a hotspot of biodiversity (if this is what you mean) is necessary.

Au. -> We reworded it L59: “the Chocoan lowland rainforest region”

L116: It is not the same domesticated lines that F1 of field captured individuals. Domesticated lines imply a number of generations under selective pressures given in captivity. This might do the difference.

Au. -> That is right. We changed “domesticated” to “laboratory raised” to avoid confusions. We wanted to show that even differences observed between wild and domesticated lines may be related to different previous acclimatization temperatures rather than exposure to captive conditions.

L129: “that thermal physiology can increase”, this sentence has not sense. Thermal physiology may vary, but not increase or decrease. Thermal critical limits may increase or decrease. The sentence is vague and should be re-written.

Au. -> We replaced “increase” with “change”.

L148: You say: “water temperature was taken at the bottom of the pond at a depth of approximately half a meter, while the tadpoles were generally on the top of the water layer where temperatures are usually higher.” This is awkwardness. If you know that tadpoles inhabit in the top of the water, why do you measure temperature at bottom?

Au. ->There are different reasons why we measured the bottom and not the top of the water. First, it is methodologically difficult to maintain the datalogger at a specific depth. Second, if loggers are put at the top they become too visible. We have lost some loggers for this reason (people are curious). But the most important reason is that to evaluate the risk of the species to suffer heat shocks we tried to deploy dataloggers at the bottom of the pond, thought to be the coolest place. Tadpoles could not select for colder pond areas and, thus, warming tolerance estimates would be more accurate.

References

Brattstrom, B.H. Thermal acclimation in Anuran amphibians as a function of latitude and altitude. Comp. Biochem. Physiol. 1968, 24, 93-111.

Lewis, C.H.R.; Richards-Zawacki, C.L.; Ibáñez, R.; Luedtke, J.; Voyles, J.; Houser, P.; Gratwicke, B. Conserving Panamanian harlequin frogs by integrating captive-breeding and research programs. Biol. Conserv. 2019, 236, 180-187.

Jensen, A.; Alemu, T.; Alemneh, T.; Pertoldi, C.; Bahrndorff, S. Thermal acclimation and adaptation across populations in a broadly distributed soil arthropod. Funct. Ecol. 2019, 33, 833-845.

Reviewer 3 Report

General Comments:

This manuscript addresses an important issue for physiological studies of rare organisms:  Can physiological values for captive populations be used to infer those of wild populations? This question is especially relevant for populations that might be sensitive to climate change or other physiological stressors, but for which too few populations remain in the wild to justify such studies on wild individuals. The authors conducted an interesting study on a single species kept in captivity for one generation, but it is unclear how generalizable the results are to other species, physiological tolerances, or more generations in captivity, despite suggestions to that effect in the Introduction.

Although the manuscript is generally well-written and study appropriate to address some aspects of the manuscript’s central question, much about the manuscript could be improved. For example, the authors should consider limiting the scope of what the study aims to accomplish in the final paragraph of the Introduction to better align with the study carried out in this manuscript. The first paragraph of the Results also should be re-worded to avoid implying acceptance of a null hypothesis, and the second paragraph of the Results should avoid stating patterns that are broadly known, and instead focus on how the thermal environment is relevant to the study organism. The Discussion would be stronger if it ended with the important take-away message from this study, rather than stating the study’s limitations and calls for further research. In general, the authors did an excellent job of pointing out the breadth of the problem, but after providing this context, the study of a single species held in captivity for one generation seemed not to reach far enough. I think this could be remedied by more carefully framing why even studies of a single species held in captivity for one generation contribute to our understanding of when and how captive populations can serve as surrogates for wild populations.

Specific comments referenced to line numbers appear below.

Specific Comments:

Line 16:                       I recommend replacing “have altered their” with “alter.”

Line 47:                       Delete “have.”

Lines 50–52:               Note that this study only evaluates whether captive-bred populations differ from wild-caught populations for studies of physiology and ecology after only one generation in captivity, and for only one species.

Line 63:                       Delete the word “to” following the word “exposed.”

Line 84:                       Replace “weighted” with “weighed.”

Line 88:                       Delete comma.

Lines 92–95:               In a hypothesis-testing paradigm (as written here), one can only reject the alternative hypothesis of a difference—one cannot accept a null hypothesis of no difference. At a minimum, I suggest deleting the word “strong” on line 92. Using a multimodel approach is a bit more nuanced. One typically uses some cutoff value below which models are determined to have little support, usually a ΔAIC of > 2 (or to be more conservative, > 7; Anderson, 2008); however, Anderson (2008) and others indicate that variables that add little explanatory power are often mistakenly interpreted as supported because they are carried by the fit of the less parameterized model (similar log-likelihood—see Table 1), but are only penalized by 2 AIC units per parameter. The authors correctly interpret little support for these variables. Nonetheless, these “pretending variables” make it difficult to state the degree of support for the null hypothesis relative to alternatives if the null is deemed to have better fit, though the principle of parsimony would suggest using the null model in these cases. I don’t doubt the authors’ conclusions of no difference, just suggest more care in wording this sentence.

Lines 96–98:               This whole paragraph seems to state a well-known, general phenomenon. The greater relevance here is how these temperatures related to the estimated CTmin and CTmax.

Figure 2:                      Because of the potential for seasonal variation in temperatures, I recommend plotting only the data (and displaying the maximum and minimum temperatures for air and water) for the dates for which both air and water temperatures were recorded.

Line 113:                     Replace “vary” with “varies.” Also, note that “methodological procedures” were not tested here. I recommend replacing “methodological procedures” with “captivity” or something similar.

Lines 116–117:           This sentence is unclear to me. Does this statement mean that there were few studies, or that of the studies conducted, few found differences?

Line 126:                     I recommend replacing “expected” with “thought.”

Line 129:                     I recommend replacing “increase” with “change” as one could imagine changes in either direction.

Line 137:                     Delete “largely.”

Lines 159–162:           This major limitation of the study matches the title well, but indicates that the results are much narrower than promised in the Introduction.

Lines 159–165:           This ending of the last paragraph of the Discussion is rather weak. I suggest highlighting the importance of demonstrating that captive-bred individuals can be used as proxies for wild populations in physiological and ecological studies at the end of the Discussion (as indicated earlier in the paragraph. Study limitations and suggestions for future research would be better placed earlier in the Discussion rather than at the power position at the end.

Literature Cited

Anderson, D. R. (2008). Model Based Inference in the Life Sciences: A Primer on Evidence. New York, NY: Springer.

Author Response

Response to Reviewer 3 Comments

This manuscript addresses an important issue for physiological studies of rare organisms:  Can physiological values for captive populations be used to infer those of wild populations? This question is especially relevant for populations that might be sensitive to climate change or other physiological stressors, but for which too few populations remain in the wild to justify such studies on wild individuals. The authors conducted an interesting study on a single species kept in captivity for one generation, but it is unclear how generalizable the results are to other species, physiological tolerances, or more generations in captivity, despite suggestions to that effect in the Introduction.

 Although the manuscript is generally well-written and study appropriate to address some aspects of the manuscript’s central question, much about the manuscript could be improved. For example, the authors should consider limiting the scope of what the study aims to accomplish in the final paragraph of the Introduction to better align with the study carried out in this manuscript. The first paragraph of the Results also should be re-worded to avoid implying acceptance of a null hypothesis, and the second paragraph of the Results should avoid stating patterns that are broadly known, and instead focus on how the thermal environment is relevant to the study organism. The Discussion would be stronger if it ended with the important take-away message from this study, rather than stating the study’s limitations and calls for further research. In general, the authors did an excellent job of pointing out the breadth of the problem, but after providing this context, the study of a single species held in captivity for one generation seemed not to reach far enough. I think this could be remedied by more carefully framing why even studies of a single species held in captivity for one generation contribute to our understanding of when and how captive populations can serve as surrogates for wild populations.

Au.: In the revised version we have included the comments and suggestions proposed. Thank you for the effort put in the manuscript. I respond to the points in the following section.

Specific Comments:

Line 16:                       I recommend replacing “have altered their” with “alter.”

Au. -> Done

Line 47:                       Delete “have.”

Au. -> Done

Lines 50–52:               Note that this study only evaluates whether captive-bred populations differ from wild-caught populations for studies of physiology and ecology after only one generation in captivity, and for only one species.

Au. -> We agree. We reworded the phrase to make the objective less bold.

 Line 63:                       Delete the word “to” following the word “exposed.”

Au. -> Done

 Line 84:                       Replace “weighted” with “weighed.”

Au. -> Done

Line 88:                       Delete comma.

Au. -> Done

 Lines 92–95:               In a hypothesis-testing paradigm (as written here), one can only reject the alternative hypothesis of a difference—one cannot accept a null hypothesis of no difference. At a minimum, I suggest deleting the word “strong” on line 92. Using a multimodel approach is a bit more nuanced. One typically uses some cutoff value below which models are determined to have little support, usually a ΔAIC of > 2 (or to be more conservative, > 7; Anderson, 2008); however, Anderson (2008) and others indicate that variables that add little explanatory power are often mistakenly interpreted as supported because they are carried by the fit of the less parameterized model (similar log-likelihood—see Table 1), but are only penalized by 2 AIC units per parameter. The authors correctly interpret little support for these variables. Nonetheless, these “pretending variables” make it difficult to state the degree of support for the null hypothesis relative to alternatives if the null is deemed to have better fit, though the principle of parsimony would suggest using the null model in these cases. I don’t doubt the authors’ conclusions of no difference, just suggest more care in wording this sentence.

 Lines 96–98:               This whole paragraph seems to state a well-known, general phenomenon. The greater relevance here is how these temperatures related to the estimated CTmin and CTmax.

Au. -> We agree. Our statement was incorrect. We made changes following the proposal of reviewer 1. L97-99: “We found no difference in critical thermal traits between wild-caught tadpoles and captive bred tadpoles of A. spurrelli. Specifically, wild-caught and captive-bred tadpoles exhibited similar CTmax whilst wild-caught and captive-bred tadpoles exhibited similar CTmin

 Figure 2:                      Because of the potential for seasonal variation in temperatures, I recommend plotting only the data (and displaying the maximum and minimum temperatures for air and water) for the dates for which both air and water temperatures were recorded.

Au. -> The temperature in Ecuador, and specifically in this locality, is quite aseasonal. We initially prepared the graph as you propose, however we believe it is more informative as shown on a daily basis. We added in L104-105: “Temperatures from Durango did not exhibit seasonal patterns; thermal variation is largely driven by diel temperature variation”.

 Line 113:                     Replace “vary” with “varies.” Also, note that “methodological procedures” were not tested here. I recommend replacing “methodological procedures” with “captivity” or something similar.

Au. -> Done

 Lines 116–117:           This sentence is unclear to me. Does this statement mean that there were few studies, or that of the studies conducted, few found differences?

Au. -> We decided to eliminate the word “Few” to avoid possible confusions.

 Line 126:                     I recommend replacing “expected” with “thought.”

Au. -> Done

Line 129:                     I recommend replacing “increase” with “change” as one could imagine changes in either direction.

Au. -> Done

 Line 137:                     Delete “largely.”

Au. -> Done

 Lines 159–162:           This major limitation of the study matches the title well, but indicates that the results are much narrower than promised in the Introduction.

 Lines 159–165:           This ending of the last paragraph of the Discussion is rather weak. I suggest highlighting the importance of demonstrating that captive-bred individuals can be used as proxies for wild populations in physiological and ecological studies at the end of the Discussion (as indicated earlier in the paragraph. Study limitations and suggestions for future research would be better placed earlier in the Discussion rather than at the power position at the end.

Au. -> We reordered and broadened the two last paragraphs. We believe the last paragraph has improved in terms of the take-away message.

 Literature Cited

Anderson, D. R. (2008). Model Based Inference in the Life Sciences: A Primer on Evidence. New York, NY: Springer.

Round 2

Reviewer 1 Report

The authors have addressed the concerns raised during the first round of reviews, and I find the revised version of the manuscript satisfactory.

Reviewer 2 Report

I appreciate the effort of the authors, but, unfortunately, their response mostly confirms my concerns. Although several of my concerns have been satisfactorily responded, others remain unsolved. Authors claim for logistic limitations and highlight that not much studies on vertebrates have been done. I recognize that logistic limitations are very important in field studies and it is very difficult to solve them. The point is if those limitations reduce the validity of the study or completely invalid it. It is probably a question of where the threshold is put.

In my opinion, this study is not valid. The main reason is that only two pairs provide all the tadpoles in captivity. Sample size is insufficient for me. The study suffers from strong pseudoreplication. Moreover, there are some flaws that I do not understand by the unnecessary. For example, field tadpoles were larger than captive tadpoles. Why? It is very simple to do that body size is similar, particularly for captive tadpoles. Data loggers were put in the bottom (where the tadpoles do not live) in order to prevent them to being stolen. I understand this trouble, but the solution is unsatisfactory: to measure temperature where the tadpoles not stay is the same as not to measure the temperature. Even worst, because we have data that might conduct to erroneous conclusions. 

Labs should be considered as a novel environment. Therefore, to say that there is or not difference between lab and field is not to say much. If thermal conditions in lab are similar to that in the field population studied, results may be similar, otherwise, results may be different. Similarly, if we compare wild populations under similar thermal conditions, results could agree, but if populations differ in their environment, thermal physiology would differ. Therefore, to achieve sound conclusions regarding the question of the authors in this study, a more elaborate design is necessary.

Reviewer 3 Report

The authors have nicely addressed my concerns with the previous version of this manuscript.